# The Mechanical and Energy Release Performance of THV-Based Reactive Materials

**DOI:** 10.3390/ma15175975

**Published:** 2022-08-29

**Authors:** Mengmeng Guo, Yanxin Wang, Haifu Wang, Jianguang Xiao

**Affiliations:** 1State Key Laboratory of Explosion Science and Technology, Beijing Institute of Technology, Beijing 100081, China; 2College of Mechatronic Engineering, North University of China, Taiyuan 030051, China; 3Science and Technology on Transient Impact Laboratory, No. 208 Research Institute of China Ordnance Industries, Beijing 102202, China

**Keywords:** THV-based reactive materials, mechanical performances, thermal analysis, reaction threshold, energy release behavior

## Abstract

A polymer of tetrafluoroethylene, hexafluoropropylene, and vinylidene fluoride- (THV) based reactive materials (RMs) was designed to improve their density and energy release efficiency. The mechanical performances, fracture mechanisms, thermal behavior, energy release behavior, and reaction energy of four types of RMs (26.5% Al/73.5% PTFE, 5.29% Al/80% W/14.71% PTFE, 62% Hf/38% THV, 88% Hf/12% THV) were systematically researched by conducting compressive tests, scanning electron microscope (SEM), differential scanning calorimeter, thermogravimetric (DSC/TG) tests and ballistic experiments. The results show that the THV-based RMs have a unique strain softening effect, whereas the PTFE-based RMs have a remarkable strain strengthening effect, which is mainly caused by the different glass transition temperatures. Thermal analysis indicates that the THV-based RMs have more than one exothermic peak because of the complex component in THV. The energy release behavior of RMs is closely related to their mechanical properties, which could dominate the fragmentation behavior of materials. The introduction of tungsten (W) particles to PTFE RMs could not only enhance the density but also elevate the reaction threshold of RMs, whereas the reaction threshold of THV-based RMs is decreased when increasing Hf particles content. As such, under current conditions, the THV-based RMs (88% Hf/12% THV) with a high density of 7.83 g/cm^3^ are adapted to release a lot of energy in thin, confined spaces.

## 1. Introduction

In order to effectively attack light armored targets, armor-piercing projectiles are usually filled with incendiary agents or explosives of a certain quality, forming armor-piercing incendiary projectiles, armor-piercing incendiary explosive projectiles, and other highly effective damage munitions [1,2]. An important feature of this kind of ammunition is that there is generally no fuze inside the warhead, which can provide more space for high-energy materials or fragile metals with high damage ability. Therefore, more efficient damage elements can be loaded to achieve a more pronounced damage effect. However, these projectiles usually explode first and then inert armor-piercing, when they impact on the target. The problem is that the incendiary explosive reaction with high damage ability only works in front of the target and could not produce a large damage effect inside the target. Therefore, people began to explore new ways of damaging light armored targets. In 2005, Daniel B. Nielson et al. [3,4] proposed, in their patent, the idea of using reactive materials enhanced projectile to achieve efficient damage to thin wall armor. Dozens of RMs formulations were designed and the performances such as overpressure, perforation, and firelight size of reactive material enhanced projectiles in spacer plates were tested. Significantly different from traditional projectiles that use inert metal to penetrate, this enhanced projectile uses reactive damage elements to pierce armor, which could be initiated and releases a large amount of chemical energy during or after its penetration process, that is, it realizes the combined effect of projectile penetration and internal explosion, thus, greatly improving the damage effect inside the target [5,6,7].

As the basic formula of fluorine polymer-based reactive material, the reactive damage element prepared from aluminum and polytetrafluoroethylene powder (Al/PTFE) through mechanical mixing, molding, and sintering process has been extensively studied by scholars and has achieved certain application results [8,9,10,11,12,13]. For example, in 2002, E. L. Baker et al. [14,15] proposed the technical concept of a unitary terminal chemical energetic blasting warhead and designed four different formulations of Al/PTFE reactive liners (inert metal type, oxygen-rich type, oxygen balance type, and oxygen deficiency type) to verify the damage effect of this unitary chemical energetic blasting warhead. The results show that, as compared to the aluminum liner, the RMs-shaped charge can not only penetrate the concrete target but also produces a stronger demolition effect due to the deflagration reaction in the concrete target, which greatly improves the damage efficiency of the shaped charge. Another example is its application in warhead fragments. Zhao Hongwei et al. studied the terminal demolition lethality of reactive fragments through experimental methods, and the results show that the reactive fragments have greatly improved the penetration of warheads, the ignition of the fuel, and the ability to detonate explosives [16].

However, the disadvantages of low Al/PTFE density (about 2.27 g/cm^3^) were exposed when the reactive material was used to replace the steel core in traditional armor-penetrating combustion projectiles to form reactive enhanced projectiles. Generally speaking, high density is an important factor to ensure the depth of armor piercing. However, the density of Al/PTFE reactive material is much lower than that of traditional steel core, resulting that its armor-piercing ability is greatly reduced, which seriously restricts the successful application of the reactive material in anti-light armor ammunition. In order to effectively improve the density of RMs, one method is to add high-density inert metal powder (such as tungsten and tantalum, etc.), into the RMs of Al/PTFE [17,18,19,20]. Since the inert metal hardly releases energy due to chemical reaction during the impact, this method reduces the release of chemical energy while improving the density. The introduction of inert metal also significantly reduces the energy release efficiency of the reactive material under the same condition, which is not conducive to the realization of efficient damage. Another method is to introduce metal oxides (such as CuO, MoO_3_, etc.), to the basic formula [21,22], which can form thermite with the Al component. Although thermite can react chemically and release a large amount of heat energy, almost no gaseous products are produced after the thermite reaction, which is not beneficial to achieving efficient damage. The ideal method is to replace the relatively light aluminum powder with higher density reactive metals, such as titanium (Ti), zirconium (Zr), hafnium (Hf), tantalum (Ta), uranium (U), etc. The reactive material composed of metal hafnium and fluoropolymer has the highest formation enthalpy, but metal hafnium has a higher activity. It is easy to react with oxygen in the air near the conventional sintering temperature of PTFE (about 380 °C), which may lead to sintering accidents. To improve the safety of Hf powder in the sintering process, fluoropolymers with lower melting and crystallization temperature can be selected, such as polyvinylidene fluoride (PVDF), tetrafluoroethylene-hexafluoropropylene copolymer (FEP), tetrafluoroethylene-hexafluoropropylene-vinylidene terpolymer (THV), etc. The fluorine content of THV 220 is 72.61%, which is the closest to the fluorine content of PTFE (76%), and it can complete melting and crystallization at about 120 °C, making it an ideal replacement material for PTFE [23]. Therefore, the RMs composed of high-density active metal powder and THV are an ideal formula for both the energy release efficiency and the density of the damage element.

Traditional PTFE-based RMs are usually prepared by a mechanical mixing–molding–sintering process, while THV products are generally large particles rather than ultra-fine powder form, so it is impossible to obtain a uniform mixture of metal powder and THV particles by traditional mechanical mixing means, the traditional preparation process is obviously not suitable for THV-based RMs. In this paper, the reactive material samples composed of metal Hf and THV were prepared by solvent evaporation and hot pressing sintering method, and the microstructure of the samples was analyzed by scanning electron microscopy. In addition, many studies have shown that the energy release of RMs is related to the fracture process of materials [24,25]. The formation of crack is related to the strain energy absorbed by the material before fracture. The larger the strain energy is, the more energy released by the material during fracture, and the easier it is to form hot spots to ignite near the crack. The strain energy absorbed by materials before fracture is closely related to their mechanical properties. Therefore, this paper intends to study the mechanical properties and fracture mechanism of THV-based RMs, and the results are conducive to a deeper understanding of the energy release mechanism of RMs.

## 2. Sample Preparation

The Hf/THV specimens were prepared by the process of solvent loss and hot pressing. The raw Hf is particle powders with the size of 10 μm, and THV 220 granules are a flexible, transparent fluoroplastic composed of tetrafluoroethylene, hexafluoropropylene, and vinylidene fluoride in the form of melt pellets, as shown in Figure 1. The preparation steps were as follows: (1) the Hf powders and THV 220 granules were poured into ethyl acetate solvent, and the concentration of THV220 should be controlled at less than 0.1 g/mL. Then, the formed solution was heated to 70–90 °C and stirred continuously until THV220 dissolved completely. (2) The solution temperature was kept at about 80 °C for distillation. To make the metal powder uniformly dispersed, continuous stirring is required during distillation until the solvent completely evaporated, resulting in a dry solid block consisting of Hf and THV 220. (3) The solid block was put into the heating mold that was placed on a universal testing machine, and the mold was heated directly to 120–160 °C in the vacuum drying box and kept warm for 30 min. Then, the pressure was loaded on the mold to 15 MPa, at a speed of 0.05 MPa/s. Finally, the sample of desired shape was obtained after 30 min of holding pressure. The density of the RMs samples fabricated by this method could reach more than 92% of theoretical maximum density (TMD). In addition, the basic formula Al/PTFE specimens were also prepared for comparison, the average size of Al powder is 10 μm as shown in Figure 1. The previous method was employed to fabricate these Al/PTFE specimens [26]. The specimen formulas and density of PTFE, THV-based RMs are listed in Table 1. The specimen density is the average density of five specimens.

Field emission scanning electron microscope (Phenom Pure, The Netherlands, desktop scanning electron microscope) was employed to investigate the microstructures of the specimens. As shown in Figure 2, Hf powders are uniformly scattered in the THV 220 matrix, but some cavities are observed in THV-based specimens, resulting in relatively low compactness for them. This may be caused by the poor compatibility between Hf and THV 220. In contrast, PTFE and Al are tightly aggregated in PTFE-based specimens, leading to high compactness of nearly 100%.

## 3. Experiment

### 3.1. Quasi-Static Compressive Test

The cylindrical specimens with a size of Φ12 mm × 12 mm were fabricated using the methods mentioned above. The static compressive test was carried out on the CMT 5105 electronic universal mechanical testing machine (MTS Industrial Systems Co., Ltd., Eden Prairie, MN, USA) at room temperature, with an initial loading strain rate of 0.8 × 10^−2^ s^−1^. The end surfaces of the specimens were polished with lubricating oil to reduce the friction with the punch of the testing machine. Under constant strain rate, three specimens were tested for each type of RMs to obtain quasi-static compressive performance curves, which are illustrated in Table 2 and Figure 6. The data in Table 2 are their average value.

### 3.2. SHPB Test

The split Hopkinson pressure bars (SHPB) system was used to investigate the dynamic compressive mechanical performance of RMs. The test system is mainly composed of one stage light gas gun subsystem, the bullet with the size of 250 × 12 mm, the compressive bar and signal collecting subsystem, etc. The different strain rates were achieved by changing the charge pressure of light gas. After the signals in the incident and transient bar are collected, the incident, reflected, and transmitted waves (*ε_i_*, *ε_r_*, and *ε_t_*, respectively), are obtained when divided by an amplification that is defined as follows: (1)$$\varepsilon_{i}=\frac{U_{i}}{C_{amp}},\;\varepsilon_{r}=\frac{U_{r}}{C_{amp}},\;\varepsilon_{i}=\frac{U_{t}}{C_{amp}},\;where\;C_{amp}=\frac{k_{1}k_{2}U_{0}}{k_{3}}$$
where *k*_1_, *k*_2_, *k*_3_ are sensitivity coefficient of strain gage, amplification of dynamic strain meter, and the amplification related to the bridge circuit. *U*_0_ is the bridge voltage. Then the strain, stress, and strain rate in the specimen could be calculated as,
(2)$$\begin{array}{l} \varepsilon_{s}=\frac{c_{0}}{l_{s}}\int (\varepsilon_{i}-\varepsilon_{r}-\varepsilon_{t})dt=-\frac{2c_{0}}{l_{s}}\int \varepsilon_{r}dt\\ \sigma_{s}=\frac{EA}{2A_{s}}\left(\varepsilon_{i}+\varepsilon_{r}+\varepsilon_{t}\right)=\frac{EA}{A_{s}}\varepsilon_{t} \end{array}$$
where *ε*_s_, *σ*_s_ are the strain and stress in the specimen, *c*_0_ is the material sound speed of the bars, *l_s_* is the length of the specimen. The dynamic mechanical performance of RMs is shown in Table 3 and Figure 7.

### 3.3. DSC/TG Test

The thermal decomposition behavior of the samples was investigated using a Mettler Toledo TGA/DSC 3+ differential scanning calorimeter and thermogravimetric analyzer. The system was programmed to heat the samples at a rate of 20 K/min from room temperature to 800 K. Sample masses of 2 mg were loaded into the sample crucible of the TGA/DSC 3+ and the sample was slightly compacted to obtain good thermal contact between the sample and the crucible. The DSC column was first evacuated using a turbo molecular drag pump and then backfilled with nitrogen. The DSC/TG experimental results of four samples are displayed in Figure 9, in which an exotherm appears as a peak while an endotherm will appear as a valley.

### 3.4. Ballistic Experiment

The closed bomb vessel with the size of Φ100 mm × 500 mm was employed to examine the energy release performance of RMs, as shown in Figure 3. The test system is mainly composed of light gas gun, speed network target, high-speed camera, pyrometer, overpressure transducer, and closed bomb vessel. RMs projectile was launched by light gas gun, and the projectile velocity was adjusted by changing the charge pressure in the gas chamber. Then, the velocity could be calculated based on the on–off signal of speed network target. The on–off signal is also the trigger signal for high-speed camera, overpressure transducer, and pyrometer. When impacting on the aluminum plate in the closed bomb vessel, the RMs projectile will be broken to be debris, by which the energy release phenomenon was induced, resulting in the abundant gas with high temperature and pressure. The released energy by the chemical reaction of RMs is also known as reaction enthalpy including the internal energy and pressure potential energy of the deflagration product. When the temperature and pressure of gas products are detected by pyrometer and overpressure transducer, respectively, the assessment of the energy release performance could be conducted.
(3)$$H=m_{p}C_{vp}\Delta T+PV$$
where *H* is the released energy by the chemical reaction of RMs, *m**_p_*, *C**_vp_*, and *T* are the mass, specific heat and temperature of gas product, *P* is overpressure, *V* is the volume of the closed bomb vessel.

#### 3.4.1. Temperature of the Gas-Phase Product

After initiation, chemical reaction occurs in RMs sample, resulting in abundant energy release in the form of internal energy of deflagration product and luminous emission that could be sensed by optical fiber probe. The optical fiber probe should be calibrated by standard light source. If the calibrated values of some wavelength were *h**_c_*, then the flash intensity could be calculated as [27,28],
(4)$$I=\frac{h_{\exp}}{h_{c}l^{2}}\cdot \frac{N_{r}(\lambda )}{2\pi (1-\cos\theta )}$$
where *h*_exp_ is the signal obtained in experiments, *l* is the distance between the object and optical fiber, *N_r_*(*λ*) is the illuminance of the standard light source, *θ* is fiber aperture angle.

After the flash intensity is achieved, the temperature of the deflagration product could be deduced based on Planck’s blackbody radiation law [29].
(5)$$I(\lambda ,T)=\varepsilon \cdot c_{1}\cdot \lambda^{-5}\cdot {\left[\exp(c_{2}/\lambda T)-1\right]}^{-1}$$
where *c*_1_ and *c*_2_ are Planck constants, *ε* is radiation coefficient, *λ* is the wavelength, *T* is the temperature of blackbody.

Because the radiation coefficient is variational in different environments, the flash intensity results of at least two wavelengths are required in general to achieve the deflagration product, by the least square method. However, the disadvantage of this method is that it requires higher experimental equipment and measurement accuracy because small changes in single-channel observations can cause large changes in radiation temperature, and the obtained radiation temperature curve is not completely reasonable. As such, a ratio method is developed to obtain more accurate results. In the current experiments, the flash intensity of four wavelengths (400 nm, 500 nm, 600 nm, 700 nm) was recorded, as illustrated in Figure 4. Then, an expression is introduced to find the temperature,
(6)S=I1I3/(I2I4)=(λ1λ3λ2λ4)−5[exp(c2λ2T)−1]·[exp(c2λ4T)−1][exp(c2λ1T)−1]·[exp(c2λ3T)−1]
where *I*_1_, *I*_2_, *I*_3_, *I*_4_ are flash intensity of the four wavelengths, respectively. The flash intensity has been obtained in the experiments, then the only unknown variable in Equation (6) is temperature *T*. From Equation (6) one can obtain the temperature of the deflagration product by numerical method. The typical temperature curves are shown in Figure 4, the results of low-velocity impact experiments could be found in Table 4.

#### 3.4.2. Overpressure in the Closed Bomb Vessel

Figure 5 shows the typical overpressure curve of RMs projectile in the closed bomb vessel. It can be seen from the figure that a pressure peak will be generated quickly after the deflagration reaction occurs, because the abundant gaseous products generated at the initial stage of the reaction sharply compress the air in the environment, resulting in a strong shock wave. As the reaction continues, the gas volume continues to increase, and shock waves continue to be generated. These shock waves undergo multiple reflections and superpositions in the closed bomb vessel, and the gas pressure tends to equalize, eventually forming the quasi-static pressure, which is the peak of the red dotted line in the figure. The impact of the projectile against the front aluminum plate of the container creates a pressure relief hole, and the pressure gradually decreases to the ambient pressure. The duration of overpressure can reach hundred milliseconds, while the temperature rise only occurs in tens of microseconds, which is mainly caused by the shielding of reaction products. On the other hand, the good sealing of closed space is also the reason for the longer duration of overpressure.

## 4. Result and Discussion

### 4.1. Mechanical Performance of RMs

From Figure 6, one can see that the fracture compression strain of fluorine polymer-based RMs is more than 2.05, except for 88% Hf/12% THV specimens. This may be attribute to the low volume fraction of THV matrix in this formula. However, the fracture strength of this formula is the highest among them, which is about 93.5 MPa. In addition, there is no distinct yield stage in the loading process for this formula, it is to say, the 88% Hf/12% THV specimens have the highest brittleness. In addition, significantly different from PTFE-based RMs, the strain softening effect is observed in THV-based RMs (Figure 6c,d) immediately after the materials yield. When the specimens are further compressed, the stress is increases with strain again, until the final rupture. The mechanisms will be discussed in detail in the following section.

Figure 7 shows the dynamic compressive curves of RMs. All specimens exhibit a remarkable strain rate enhancement effect. The higher the strain rate, the stronger the materials. The stress of PTFE-based RMs is generally increasing with a strain before failure. However, the stress of THV-based RMs shows a downward trend with strain after the yield point, indicating that the THV-based RMs are strain-softening materials, whereas the PTFE-based RMs are strain-hardening materials.

### 4.2. The Fracture Mechanism

Field emission scanning electron microscope (Phenom Pure, desktop scanning electron microscope) was employed to examine the micromorphology of the surface of the specimen after compression of these RMs, as shown in Figure 8. When compression was loaded in the vertical direction, the materials would expand in the horizontal direction because of the Poisson effect, leading to the metal particles being separated from the fluorine polymer matrix (Figure 8a,b). From Figure 8, one can see that the THV presents a more remarkable orientation near the microcosmic interface with metal than PTFE. This may be mainly caused by the different glass transition temperatures of them, which has a significant effect on the movement of disordered chains of molecules in RMs. If the ambient temperature remains under the glass transition temperature, the movement of the disordered chains of molecules is restricted, and relative motion between them could be only found in a small range (Figure 8a). In this condition, the stress is increasing with the deformation of the fluorine polymer matrix until the fracture. The ambient temperature of the compression test is about 25 °C, which is far below the glass transition temperature of PTFE (115 °C). Therefore, the PTFE-based RMs are “frozen” in this case. However, the glass transition temperature of THV is 5 °C, which is below the test temperature. In this case, the disordered chains of molecules in RMs will be “unfrozen” so that the relative motion between them is promoted to a great extent under loading. For the deformation of RMs during the period of movement of disordered chains of molecules, the materials are apt to present the characteristics of the fluid, resulting in the non-continuously increased stress suffered by them.

### 4.3. Thermal Behavior under DSC/TG Tests

The TG/DSC curves of Al/PTFE are depicted in Figure 9a. It can be determined that the endothermic peak A is the melting temperature of PTFE, and the endothermic peak C is the melting temperature of Al. The endothermic peak B covers a temperature range from 530 °C to 601 °C, at the same time, the mass of the samples decreases according to the TG curve, indicating that the small gas molecule is produced from the decomposition of the PTFE matrix. Generally, violent exothermic reactions between the small gas molecules and Al particles will immediately occur after the decomposition of PTFE [5,30]. However, violent exothermic reactions were not found before the Al particles are melted in this experiment. For mechanism consideration, the Al particles are often coated with a hard layer of alumina, which could be broken by volume expansion from the melted Al [31]. Then, a violent exothermic reaction occurs when small gas molecules meet fine aluminum, leading to the formation of the exothermic peak D in the DSC curve. When W particles are introduced to the formula, violent exothermic peak B occurs immediately after the decomposition of PTFE, which could be corresponding to the reaction between W particles and small gas molecules, as shown in Figure 9b. As such, it could be inferred that the rupture of the alumina layer outside the Al core plays an important part in the onset of the whole chemical reaction of Al/PTFE RMs. If the alumina layer is ruptured under the impact load, the chemical reaction will bring forward the decomposition temperature of PTFE.

The TG/DSC curves of Hf/THV are shown in Figure 9c,d. The endothermic peak A is the melting endothermic peak of THV. Significantly different from Al/PTFE, two exothermic peaks (B and C) are observed in the DSC curves, both of which are accompanied by a decrease in mass (Figure 9c). The first exothermic peak B leads to a 17.9% drop in mass, which is consistent with the content of TFE (17.40%) in THV. The subsequent exothermic peak may be associated with the reaction caused by the HFP and VDF. In addition, from Figure 9b,d, one can find that the weight increase in the TG curve may be caused by the reaction product’s desublimation in the crucible lid. A closer look shows that there is also an uptrend in the late TG curve in Figure 9a and the middle part of the TG curve in Figure 9c. However, due to the higher content of PTFE or THV in a and c, the weight increased by the reaction product desublimation was much smaller than the weight lost by the reaction, so the uptrend of the TG curve was not obvious. For Figure 9b,d, the content of PTFE or THV was less, and the end of the reaction was earlier, so the weight increase caused by the desublimation of reaction products was more obvious. Secondly, due to the desublimation of exothermic heat, the formation of exothermic peaks was accompanied by the rise of the TG curve.

The chemical reaction processes could be summarized as follows. For Al/PTFE and Al/W/PTFE, the possible chemical reaction could be:$$\begin{array}{l} \left(-C_{2}F_{4}-\right)n\to nC_{2}F_{4}\\ 4Al+3C_{2}F_{4}\to 4AlF_{3}+6C\\ W+C_{2}F_{4}\to WF_{4}+2C \end{array}$$

For Hf/THV, it could be:$$\begin{array}{l} {\left(-C_{2}F_{4}-\right)}_{n}{\left(-C_{3}F_{6}-\right)}_{m}{\left(-CH_{2}CF_{2}-\right)}_{l}\to nC_{2}F_{4}+mC_{3}F_{6}+lCH_{2}CF\\ Hf+C_{2}F_{4}\to HfF_{4}+2C\\ 3Hf+2C_{3}F_{6}\to 3HfF_{4}+6C\\ Hf+2CH_{2}CF_{2}\to HfF_{4}+4C+2H_{2} \end{array}$$

### 4.4. The Energy Release Behavior of RMs

The energy release process in the closed vessel with the front aluminum plate of 2 mm and the back steel plate of 10 mm captured by the high-speed camera is shown in Figure 10. For the basic formula with the impact velocity of 635 m/s, as illustrated in Figure 10a, violent chemical reactions in the 26.5% Al/73.5% PTFE RMs projectile were observed in the front of the aluminum plate, whereas the chemical reactions became a little weaker in the back of the aluminum plate. Subsequently, more violent chemical reactions were induced by the rigid steel plate, which did not stop until 89.28 ms. When the impact velocity was increased to 1880 m/s (Figure 10b), the basic formula RMs projectile released much more energy at the front and back of the first aluminum plate as compared to that with the velocity of 635 m/s, and the chemical reaction induced by the steel plate became less. This is to say, a large part of the energy is released outside the closed vessel with a higher impact velocity. In addition, the whole reaction time became only 9.44 ms, indicating that the rate of a chemical reaction is highly dependent on the impact conditions, including projectile material and velocity.

When introducing tungsten particles to the basic formula, a little chemical reaction of RMs projectile with the impact velocity of 464 m/s was found at the front or back of the aluminum plate, and the obvious chemical reaction was not found until it impacted the second steel plate. Significantly different from the basic formula, the chemical reaction duration was prolonged to 172.21 ms, which may be caused by the chemical reaction with a low rate between tungsten or aluminum and oxygen in the air. When the impact velocity was increased to 1150 m/s, a violent chemical reaction was observed at the front and back of the first aluminum plate. Subsequently, the residual penetrator after the first plate produced more reaction when impacting the second steel plate, the duration was reduced to 92 ms.

For the 62% Hf/38% THV RMs projectile, the energy release pictures are illustrated in Figure 10e,f. When the impact velocity is 450 m/s, a chemical reaction only occurred before the second steel plate. When the impact velocity is increased to 1600 m/s, the chemical reactions were at a competitive level before the aluminum and steel plate, and the reactions were fast, with a duration of about 6.64 ms. When increasing the Hf content to 88%, the RMs produced a remarkable chemical reaction at the back of the aluminum plate. Extremely fragmentation of RMs projectile could be concluded according to the flame pattern in Figure 10g. When impacting the second steel plate, a more violent chemical reaction occurred, and the duration was about 209.86 ms, which could be caused by the chemical reaction with a low rate between the hafnium and oxygen in the air. When the impact velocity was increased to 1100 m/s, a certain chemical reaction occurred before the first aluminum plate, whereas most of the energy was released inside the closed vessel, which met the original requirement of the RMs projectile.

From the above analysis, one can conclude that the energy release process of RMs projectile is mainly dependent on the mechanical characteristics of RMs and the impact condition. The mechanical characteristics of RMs are mainly reflected in the stress–strain curve (Figure 6 and Figure 7). The strain energy of the RMs can be obtained by integrating the corresponding stress–strain curve. Integrating the function of curves in Figure 7, the result of the integration with a strain rate of about 4000 S^−1^ is as follows: (a) 14.8588 KJ, (b) 23.8584 KJ, (c) 32.2153 KJ, (d) 4.40197 KJ. As can be seen from the results of the integration, when increasing the Hf content to 88%, the THV-based RMs become easier to be fragmented. The introduction of tungsten (W) particles to PTFE RMs make RMs not easy to be fragmented. In fact, fragmentation of RMs projectile is likely the prerequisite condition for ignition because of the general non-self-sustaining reaction in RMs. If it is the fact, the reaction process is a particle burning process in essence. The addition of W particles increases the density of the material and thus improves its penetration ability so that the strain is smaller during perforating the first Al plate than that suffered by the basic formula, leading to a smaller amount of fragmentation to release energy before and after the first Al plate (Figure 10c). The reaction threshold of THV-based RMs is higher because of their unique strain softening effect, which is not conducive to breakage. However, the reaction threshold of THV-based RMs is decreased when increasing the Hf content to 88%. This is because the high content of Hf powder improves the fracture strength of the material and increases the energy released near the crack after the fracture of the material to generate a hot spot at a higher temperature, thus making the RMs more prone to ignition reaction. In conclusion, the introduction of tungsten (W) particles to PTFE RMs could not only enhance the density but also elevate the reaction threshold of RMs, whereas the reaction threshold of THV-based RMs is decreased when increasing Hf particle content to achieve an equivalent density of RMs projectile. As such, the THV-based RMs with high density is adapted to release a lot of energy in thin, confined spaces. This has important reference significance to the design of the RMs warhead.

### 4.5. The Total Energy Release of THV-Based RMs

Since deflagration reaction time is very short, the heat radiated by deflagration gas products to the environment is ignored, and it is considered that there is no heat exchange between deflagration products and the external environment. Therefore, the temperature of the gas in the closed bomb vessel can be considered as its radiation temperature. Then, the final released energy by the reactive material projectile can be calculated according to Equation (3). As listed in Table 4, for the basic formula, the energy release efficiency is 26.69% when the impact velocity is 635 m/s. When the speed was increased to 1880 m/s, the energy release efficiency decreased to 24.90%. On one hand, the increase in velocity will increase the friction between the projectile and the gun barrel, resulting in the obvious temperature rising phenomenon of the projectile after leaving the muzzle. The temperature rising phenomenon under high strain rate load will make part of the RMs react outside the closed bomb vessel. On the other hand, when the velocity is increased, the impact stress between the reactive projectile and the first aluminum plate is higher, so that more RMs are broken and react in front of the aluminum plate to release energy. The result of both actions is that the RMs release a large amount of energy outside the closed container. So, even though the velocity is higher, the energy released inside the container is lower. When tungsten powder is introduced to the basic formula, the density of the projectile is close to that of steel, so the degree of breakage of the reactive projectile before the first aluminum plate is reduced, and the energy release efficiency of the reactive projectile in the container is improved. When the velocity level is the same, the energy release efficiency of THV-based RMs is lower than that of PTFE-based RMs, indicating that the reaction threshold of THV-based RMs is higher. The main reason is that the toughness of THV-based RMs is better than that of PTFE-based RMs. Under the same impact conditions, the degree of breakage of THV-based RMs is lower than that of PTFE-based RMs, and the reaction degree is lower, resulting in lower energy release efficiency. After further increasing the proportion of hafnium powder, the strength and brittleness of THV-based RMs are improved, and the difficulty of crushing is significantly reduced. Therefore, under the same impact conditions, the fragmentation degree and energy release efficiency are also improved, reaching 121.15% at high speed. At high speed, the energy release efficiency of the reactive projectile of various formulations is more than 100%, and the highest is 135.32. The reasons for this overestimation are twofold: (1) the kinetic energy of the projectile is not considered; (2) the temperature in the closed bomb vessel is constantly changing as the reaction progresses; however, the highest temperature is used to calculate the internal energy, resulting in the overestimation for the internal energy listed in Table 4.

## 5. Conclusions

In this work, four types of PTFE- and THV-based RMs were prepared. The mechanical performance and reactive characteristics of the materials were systematically investigated through quasi-static compressive tests, SHPB tests, SEM investigations, DSC/TG tests, and ballistic experiments. Combined with the above analysis, the impact-induced energy release process of RMs could be summarized as follows: (1) the fragmentation of RMs samples; (2) the product of small gas molecules by the decomposition of fluoropolymer matrix under impact loading (impact-induced hot spot or impact-induced fracture); (3) exposure of reactive metal to small gas molecule atmospheres; (4) burning process of the fragmentized composite particle. The main conclusions can be drawn as follows: (1)For the compression tests with quasi-static strain rate, the THV-based RMs have a unique strain softening effect whereas the PTFE-based RMs have a remarkable strain strengthening effect, that is, the stress decreases with the increase in strain after the materials yield. This phenomenon is mainly caused by the different glass transition temperatures of them. The glass transition temperature of THV is 5 °C, which is below the test temperature. In this case, the disordered chains of molecules in RMs will be “unfrozen” so that the relative motion between them is promoted to a great extent under loading, leading to their unique strain softening effect.(2)Thermal analysis indicates that the THV-based RMs have more than one exothermic peak because of the complex component in THV. The first exothermic peak leads to a 17.9% drop in mass, which is consistent with the content of TFE (17.40%) in THV. The subsequent exothermic peak may be associated with exothermic reaction caused by the HFP and VDF. In addition, the rupture of the alumina layer outside the Al core plays an important part in the onset of the whole chemical reaction of Al/PTFE RMs. If the alumina layer is ruptured under the impact load, a chemical reaction will bring forward the decomposition temperature of PTFE. Additionally, the increase in the TG curve is caused by the reaction product’s desublimation in the crucible lid.(3)The reaction threshold is closely related to the mechanical characteristics of RMs. The introduction of tungsten (W) particles to PTFE RMs could not only enhance the density but also elevate the reaction threshold of RMs, whereas the reaction threshold of THV-based RMs is decreased when increasing Hf particles content to achieve an equivalent density of RMs projectile. This is because the high content of Hf powder makes it easier for the RMs to be fragmented and it increases the energy released near the crack after the fracture of the material to generate a hot spot at a higher temperature, thus making the RMs more prone to ignition reaction. However, the introduction of tungsten (W) particles to PTFE RMs make RMs not easy to be fragmented. As such, under current conditions, the THV-based RMs (88% Hf/12% THV) with a high density of 7.83 g/cm^3^ are adapted to release a lot of energy in thin, confined spaces.

## Figures and Tables

**Figure 1 materials-15-05975-f001:**
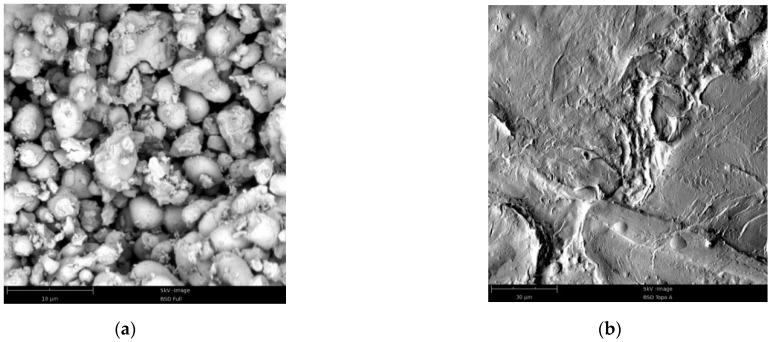
The scanning electron microscope (SEM) images of the raw materials: (**a**) Hf powder; (**b**) THV 220 granules; (**c**) Al powder; (**d**) PTFE powder.

**Figure 2 materials-15-05975-f002:**
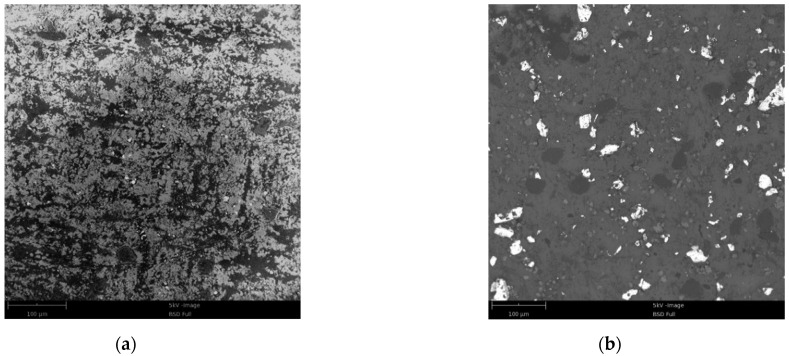
Microstructures of RMs specimens: (**a**) 26.5% Al/73.5% PTFE; (**b**) 5.29% Al/80% W/14.71% PTFE; (**c**) 62% Hf/38% THV; (**d**) 88% Hf/12% THV.

**Figure 3 materials-15-05975-f003:**
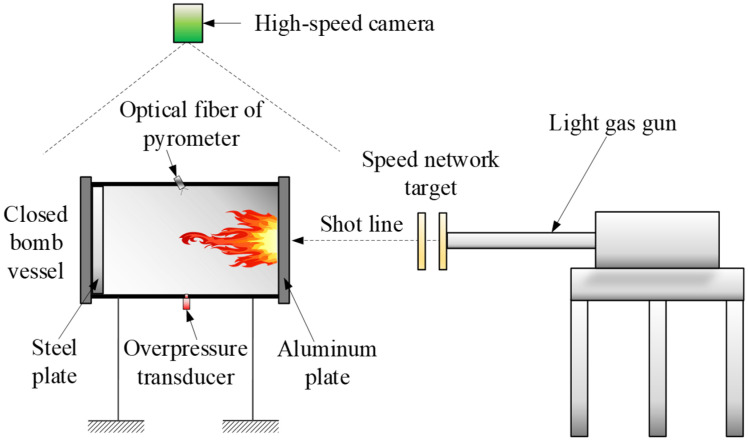
Test system of the energy release performance of RMs.

**Figure 4 materials-15-05975-f004:**
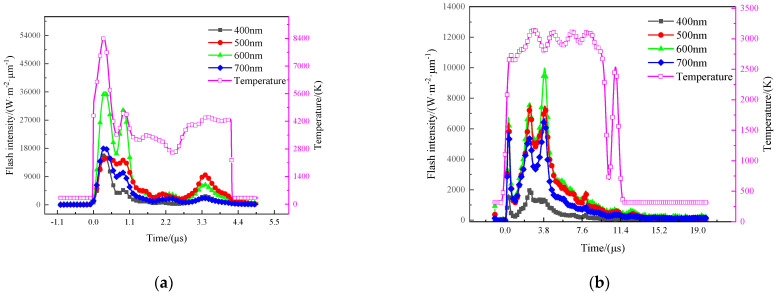
Electrical signal recorded by transient pyrometer and temperature of deflagration flame: (**a**) 62% Hf/38% THV−1500 m/s (**b**) 88% Hf/12% THV−1100 m/s (**c**) 26.5% Al/73.5% PTFE−1880 m/s (**d**) 5.29% Al/80% W/14.71% PTFE−1150 m/s.

**Figure 5 materials-15-05975-f005:**
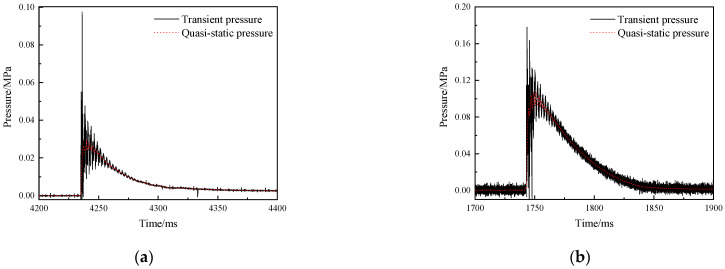
Typical overpressure versus time in the closed bomb vessel: (**a**) 26.5% Al/73.5% PTFE−1880 m/s (**b**) 5.29% Al/80% W/14.71% PTFE−1150 m/s (**c**) 62% Hf/38% THV−1500 m/s (**d**) 88% Hf/12% THV−1100 m/s.

**Figure 6 materials-15-05975-f006:**
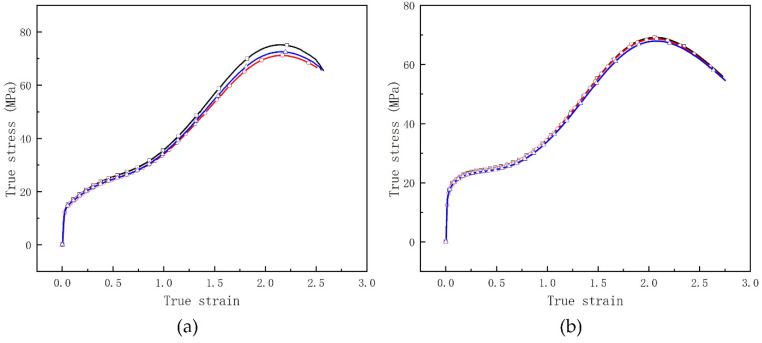
Quasi-static compressive performance curves of RMs: (**a**) 26.5% Al/73.5% PTFE (**b**) 5.29% Al/80% W/14.71% PTFE (**c**) 62% Hf/38% THV (**d**) 88% Hf/12% THV.

**Figure 7 materials-15-05975-f007:**
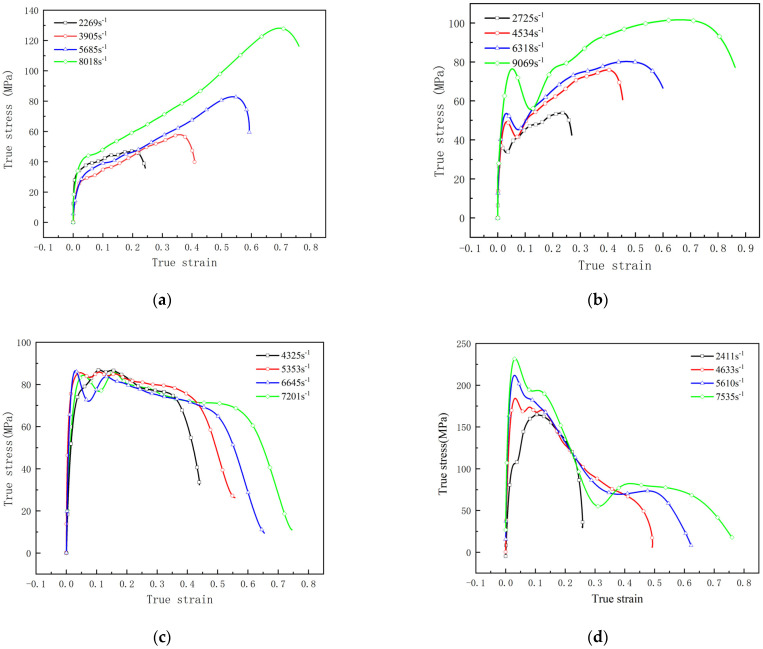
Dynamic compressive curves of RMs: (**a**) 26.5% Al/73.5% PTFE (**b**) 5.29% Al/80% W/14.71% PTFE (**c**) 62% Hf/38% THV (**d**) 88% Hf/12% THV.

**Figure 8 materials-15-05975-f008:**
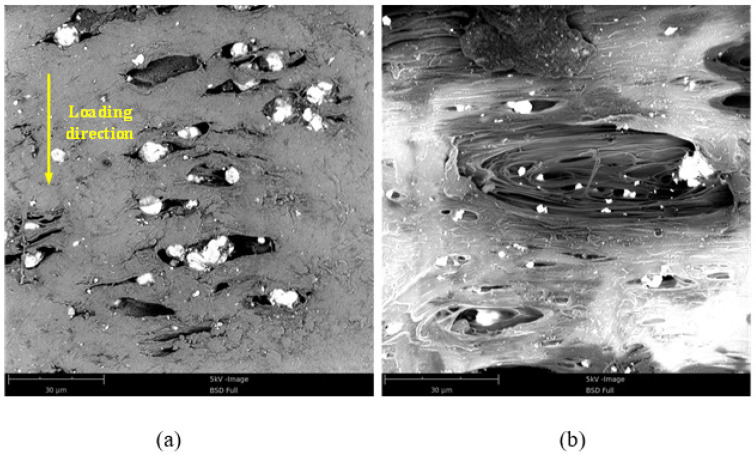
Micromorphology of the surface of specimen after compression: (**a**) PTFE-based RMs; (**b**) THV-based RMs. The surface is the lateral surface of the sample.

**Figure 9 materials-15-05975-f009:**
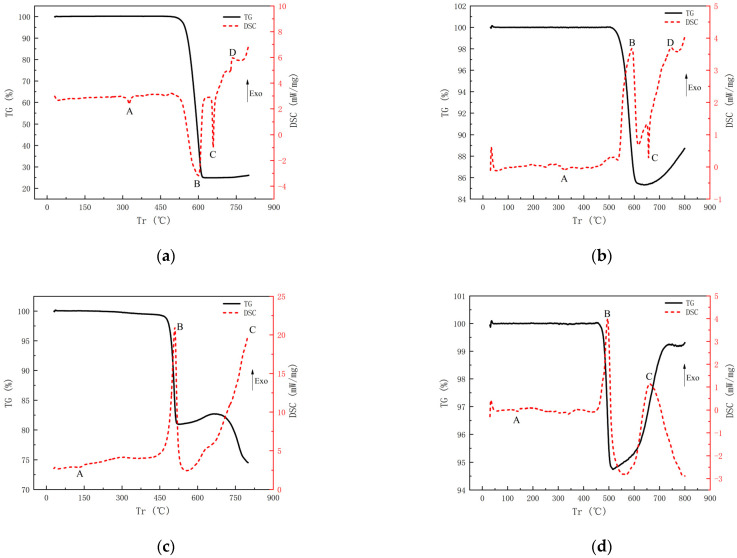
DSC/TG results of RMs: (**a**) 26.5% Al/73.5% PTFE (**b**) 5.29% Al/80% W/14.71% PTFE (**c**) 62% Hf/38% THV (**d**) 88% Hf/12% THV.

**Figure 10 materials-15-05975-f010:**
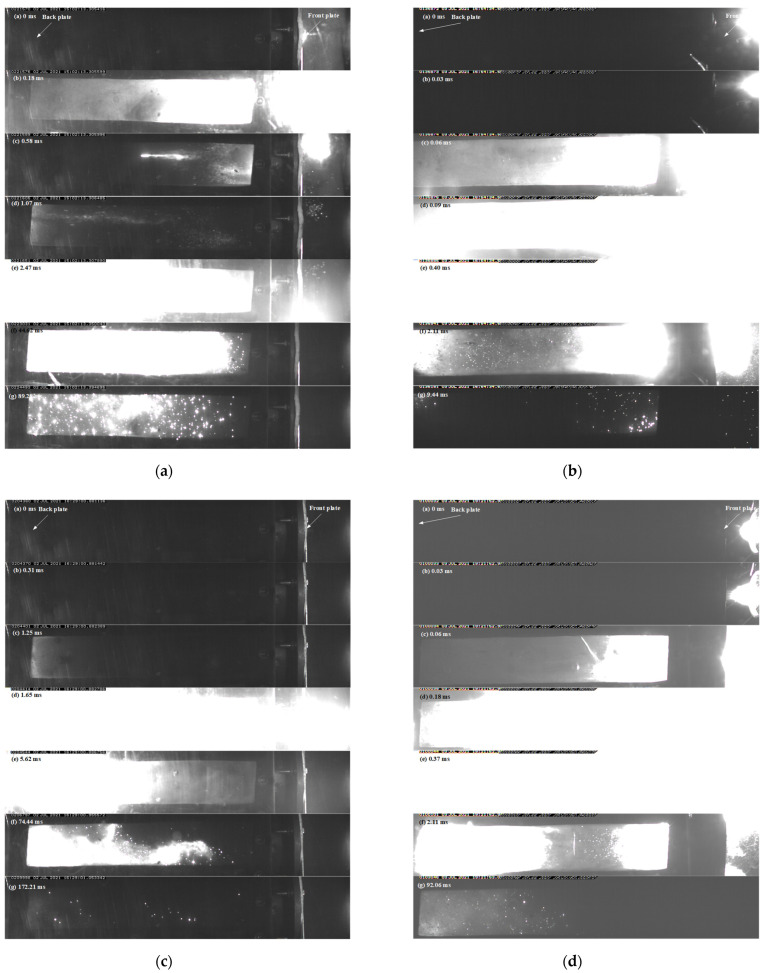
Energy release process of RMs: (**a**) 26.5% Al/73.5% PTFE-635 m/s (**b**) 26.5% Al/73.5% PTFE-1880 m/s (**c**) 5.29% Al/80% W/14.71%PTFE-464 m/s (**d**) 5.29% Al/80% W/14.71% PTFE-1150 m/s (**e**) 62% Hf/38% THV-450 m/s (**f**) 62% Hf/38% THV-1500 m/s (**g**) 88% Hf/12% THV-480 m/s (**h**) 88% Hf/12% THV-1100 m/s.

**Table 1 materials-15-05975-t001:** The specimen formulas and theoretical energy content.

NO.	Specimen Formula	Theoretical Density (g/cm^3^)	Specimen Density (g/cm^3^)	Compactness
1	26.5% Al/73.5% PTFE	2.31	2.31	100%
2	5.29% Al/80% W/14.71% PTFE	7.73	7.73	100%
3	62% Hf/38% THV	4.14	3.93	94.93%
4	88% Hf/12% THV	7.83	7.28	92.98%

**Table 2 materials-15-05975-t002:** Quasi-static compressive performance of RMs.

No.	Elasticity Modulus (MPa)	Yield Stress (MPa)	Yield Strain	Compressive Strength (MPa)	Failure Strain
1	728	13.30	0.0322	72.57	2.14
2	661	16.89	0.0308	68.60	2.05
3	665	28.85	0.1205	55.12	2.18
4	3550	79.39	0.0257	98.17	0.06

**Table 3 materials-15-05975-t003:** Dynamic compressive performance of RMs.

Formula	Strain Rate (s^−1^)	Yield Stress (MPa)	Yield Strain	Compressive Strength (MPa)	Failure Strain
26.5% Al/73.5% PTFE	2269	30.14	0.0173	47.21	0.20
	3905	21.44	0.0122	57.88	0.36
	5685	23.04	0.0165	82.91	0.54
	8081	34.65	0.0161	128.15	0.69
5.29% Al/80% W/14.71% PTFE	2725	36.05	0.0154	53.84	0.23
	4534	49.30	0.0337	75.90	0.40
	6318	53.52	0.0322	80.22	0.47
	9069	76.12	0.0479	101.65	0.66
62% Hf/38% THV220	4325	74.03	0.0374	86.85	0.35
	5353	85.56	0.0486	85.56	0.39
	6645	86.49	0.0302	86.49	0.49
	7201	84.32	0.0561	85.79	0.56
88% Hf/12% THV220	2411	103.49	0.0241	164.71	0.11
	4633	184.20	0.0321	184.20	0.03
	5610	211.79	0.0301	211.79	0.03
	7535	231.94	0.0318	231.94	0.03

**Table 4 materials-15-05975-t004:** The performance of energy release by RMs projectile.

Formula	Density (g/cm^3^)	Velocity (m/s)	Over-Pressure (Mpa)	Temper-ature (K)	Pressure Potential Energy (kJ/g)	Internal Energy (kJ/g)	Energy Content (kJ/g)	Efficiency (%)	Duration (ms)
1	2.42	635	0.128	2514	0.50	3.41	14.64	26.69	89.28
1	2.27	1880	0.0289	2169	0.50	3.15	14.64	24.90	9.44
2	7.92	464	0.089	2929	0.11	0.71	4.34	18.86	172.21
2	7.87	1150	0.106	2261	0.53	5.09	4.34	129.33	92.06
3	3.94	450	0.031	-	0.08	-	10.03	-	4.92
3	3.97	1500	0.216	4721	1.77	11.80	10.03	135.32	79.63
4	7.45	480	0.108	4664	0.13	1.99	6.97	30.42	211.30
4	7.24	1100	0.179	3140	0.88	7.57	6.97	121.15	100.67

## Data Availability

The data that support the findings of this study are available from the corresponding author upon reasonable request.

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
