# Peer review of "The Mechanical and Energy Release Performance of THV-Based Reactive Materials"

_materials, 2022, doi:10.3390/ma15175975_

Round 1

Reviewer 1 Report

An interesting study. Please, read my notes and suggestion. Also, point out that there is no statistically overview of the result. Maybe that will be done in a next paper, and this is the start of the research.

I did suggestions. Maybe they are useful.

Author Response

Dear editor,

Thanks very much for your attention on our manuscript (materials-1838343). We have made a careful revision on the revised manuscript according to the comments, and the revisions made to the manuscript have been marked up using the “Track Changes” function. The modification details are as follows:

Reviewer 1:

  1. 'tests' has been omittedin line 17.
  2. ‘unique strain softening effect whereas ‘has been corrected for‘unique strain softening effect, whereas’ in line 19.
  3. ‘introduce’ has been corrected for ‘introduction’ in line 23.
  4. ‘the density but also’ has been corrected for ‘the density, but also’ in line 24.
  5. ‘better’ has been corrected for ‘more pronounced’ in line 38.
  6. ‘inert armor-piercing when’has been corrected for ‘inert armor-piercing, when’ in line 39.
  7. ‘proposed in their patent the idea’ has been corrected for‘proposed, in their patent, the idea’ in line 43.
  8. ‘thus greatly improving’ has been corrected for‘thus, greatly improving’ in line 51.
  9. ‘[8]-[13].For example’ has been corrected for ‘[8]-[13]. For example’ in line 55.
  10. ‘[14],[15]proposed’ has been corrected for‘[14],[15] proposed’ in line 56.
  11. ‘compared with’ has been corrected for‘as compared to’ in line 60.
  12. ‘produce’ has been corrected for‘produces’ in line 61.
  13. ‘g/cm3’ has been corrected for‘g/cm3’ in line 68.
  14. ‘resulting’ has been corrected for‘resulting that’ in line 73.
  15. ‘MoO3’has been corrected for ‘MoO3’ in line 81.
  16. ‘Titanium Ti, zirconium Zr, hafnium Hf, tantalum Ta, Uranium U’ has been corrected for‘Titanium (Ti), zirconium (Zr), hafnium (Hf), tantalum (Ta), Uranium (U)’ in line 86-87.
  17. ‘is closest to’ has been corrected for‘is the closest to’ in line 94.
  18. ‘its’ has been corrected for‘their’ in line 111.
  19. ‘are’ has been corrected for‘is’ in line 116.
  20. We heated the mold directly to 120~160 ℃in the vacuum drying box, the content has been added in line 127.
  21. ‘to 15 MPa,at a speed of’ has been corrected for‘to 15 MPa, at a speed of’ in line 128.
  22. We measured five specimens to conclude ‘The density of the RMs samples fabricated by this method could reach more than 92% of theoretical maximum density (TMD).’
  23. ‘Table. 1.’ has been corrected for‘Table 1.’ in line 134. And we have corrected the same errors in the document.
  24. The Fig. 1 is a view of the sample as it is and the sample wasn’t goldened. And the problem of pictures’ scale is not visible, it is caused by the equipment.
  25. ‘Formula’ has been corrected for‘Specimen formula’ in Table 1.
  26. We re-arranged the Table 1 to have one line for the material ‘5.29%Al/80%W/14.71%PTFE’.
  27. The specimen density is the average density of five specimens in Table 1.
  28. We have three specimen for each material, under the same conditions. The data in Table 2 are their average value. We did the compression test under constant strain rate. In this regard, we have made amendments in lines160-162.
  29. We use subscript for εi, εrand εt in line 173. And we checked all the document and corrected such errors.
  30. we move Fig. 5 next to the explanations, at page 10.
  31. ‘Φ100 ×500 mm’ has been corrected for‘Φ100 mm×500 mm’ in line 205.
  32. ‘will be broken to be debris by which’ has been corrected for‘will be broken to be debris, by which’ in line 213.
  33. We added the introduction for the formula in line 220.
  34. We used subscript in line 231,line 234 and line 235.
  35. ‘400nm, 500nm, 600nm, 700nm’ has been corrected for ‘400 nm, 500 nm, 600 nm, 700 nm’ in line 244.
  36. ‘variate’ has been corrected for ‘variable’ in line 247.
  37. ‘the temperature rise is only tens of microseconds’ has been corrected for‘the temperature rise only occurs for tens of microseconds’ in line 268.
  38. This is a typical test,we compare the test results with those in other literatures, and the results are similar. Therefore, the test data is reliable. When comparing the energy release of different formulations of reactive materials, the speed should be the same or relatively close. Due to the limited test equipment and funds, the same impact speed is not obtained.(Original Figure 8, now Figure 5)
  39. ‘From Fig. 3 one can see’ has been corrected for ‘From Fig. 6, one can see’ in line 278.
  40. ‘is highest’ has been corrected for ‘is the highest’ in line 281.
  41. ‘with strain again until the final rupture’ has been corrected for ‘with strain again, until the final rupture’ in line 286.
  42. We added (a) and (b) under the images in Fig. 9 (now is Fig. 8). And the surface is the lateral surface of the sample.
  43. 'leading to' has been omitted in line 342.
  44. We made changes to the sentence beginning with ‘If the alumina’ in line 347-349.
  45. ‘from Fig. 5b and d one can find that the weight increase of the TG curve which’ has been corrected for ‘from Fig. 9b and d, one can find that the weight increase of the TG curve, which’ in line 356.
  46. ‘Subsequently more violent chemical reactions’ has been corrected for ‘Subsequently, more violent chemical reactions’ in line 383.
  47. ‘aluminum plate compared with that’ has been corrected for ‘aluminum plate as compared to that’ in line 386.
  48. ‘impact loading’ has been corrected for ‘impact conditions, including projectile material and velocity’ in lines390-391.
  49. 'the' has been omitted in line 397.
  50. ‘Subsequently the residual’has been corrected for ‘Subsequently, the residual’ in line 399.
  51. ‘were fast with the duration’ has been corrected for ‘were fast, with the duration’ in line 406.
  52. ‘From the above analysis one can’ has been corrected for ‘From the above analysis, one can’ in line 424.
  53. ‘Integrate the curves in Fig. 4, The result’ has been corrected for ‘Integrating the curves in Fig. 7, the result’ in line 428.
  54. ‘4000S-1’ has been corrected for ‘4000S-1’ in lines429.
  55. ‘(a)14.8588KJ, (b)23.8584KJ, (c)32.2153KJ, (d)4.40197KJ.’ has been corrected for ‘(a)14.8588 KJ, (b)23.8584 KJ, (c)32.2153 KJ, (d)4.40197 KJ.’ in line 429-430.
  56. The RMs is the ‘reactive materials’ abbreviation, so use ‘make’ in line 432.
  57. Table 1 is the average density, while Table 4 is the actual density of the sample. So vales of Table 4 are different from Table 1.
  58. The results show that the over-pressure so small in Table 4.
  59. ‘tests’ has been corrected for ‘investigations’ in line 490.
  60. ‘Ballistic’has been corrected for ‘ballistic’ in line 491.
  61. We added ‘For the compression tests with quasi-static strain rate, the’ in line 497.
  62. We made changes to the sentence beginning with ‘If the alumina’ in lines510-512.
  63. ‘The introduce of tungsten (W) particles to PTFE RMs could not only enhance the density but also’ has been corrected for ‘The introduction of tungsten (W) particles to PTFE RMs could not only enhance the density, but also’ in line 515-516.
  64. ‘This is because the high content of Hf powder make it easier for the RMs to be fragmented and increases the energy released’ has been corrected for ‘This is because the high content of Hf powder makes easier for the RMs to be fragmented and it increases the energy released’ in line 518-519.
  65. ‘However, the introduce of tungsten (W) particles to PTFE RMs make RMs not easy to be fragmented.As’ has been corrected for ‘However, the introduction of tungsten (W) particles to PTFE RMs make RMs not easy to be fragmented. As’ in line 521.

Reviewer 2 Report

Research by Guo et al considers new reactive materials for deflagrating projectiles with high thermal release efficiency.

The topic is certainly interesting in ballistic applications but the writing of the manuscript is to be revised both for the presence of typos and for a poor organization in the description of the contents.

Specifically, the manuscript anticipates, also with the aid of graphs, part of the results obtained in paragraph 3 which, on the other hand, must indicate only the characterization techniques and test conditions adopted. Therefore, the authors are asked to significantly revise the organization of the contents in paragraphs 3 and 4 as expected from the reading of a typical research paper.

 Moreover, below are some points to be reviewed.

Lines 69-73: the sentence starting with "Because high density ..." is long and difficult to understand after a first reading. It is advisable to break this sentence to improve its comprehensibility.

Line 123: replace “consists” with “consisting”.

Line 124: Check the sentence starting with "Put the solid ...". Probably replace "that placed" with "that is placed".

Lines 207-209: the sentence starting with "Besides" is not clear. Please check and rephrase.

Line 242: please replace the word "variate" with "variable".

Lines 303-304: The sentence starting with "Therefore" is incomplete. Please check and correct.

Lines 315-318: the sentence starting with "The endothermic peak" is convoluted. Please rephrase and explain the contents better.

Lines 323-324: Authors write "leading to leading". Please revise to eliminate the almost consecutive repetition of the word "leading".

Lines 329-330: the sentence starting with "If the alumina layer ..." is incomplete. Please check and correct.

Line 336: the adjective "exothermic" before the word "reaction" is superfluous. Please cancel it.

Line 361: The text reads "with that with". Please reprase to eliminate this repeat.

Line 398: correct "concluded" with "conclude".

Line 402: The sentence starting with "Integrate ..." is incomplete. It probably derives from a "cut and paste" operation. Please check and correct.

Line 22, line 406 and line 488: replace the word "introduce" with "introduction".

Lines 483-485: the sentence starting with "If the alumina ..." is incomplete. Please check and correct.

 In light of the above considerations, MINOR revision are firmly recommended.

Author Response

Thanks very much for your attention on our manuscript (materials-1838343). We have made a careful revision on the revised manuscript according to the comments, and the revisions made to the manuscript have been marked up using the “Track Changes” function. The modification details are as follows:

  1. Comment::Specifically, the manuscript anticipates, also with the aid of graphs, part of the results obtained in paragraph 3 which, on the other hand, must indicate only the characterization techniques and test conditions adopted. Therefore, the authors are asked to significantly revise the organization of the contents in paragraphs 3 and 4 as expected from the reading of a typical research paper.

Response: For the convenience of the reader, we have moved the figures of mechanical and thermal behavior (figure 3. figure 4 and figure 5) from paragraph 3 to paragraph 4. Now they are figure 6, figure 7 and figure 9.

  1. Comment::Lines 69-73: the sentence starting with "Because high density ..." is long and difficult to understand after a first reading. It is advisable to break this sentence to improve its comprehensibility.

Response: The sentence starting with "Because high density ..." has been modificated for ‘Generally speaking, high density is an important factor to ensure the depth of armor-piercing. However, the density of Al/PTFE reactive material is much lower than that of traditional steel core, resulting that its armor-piercing ability is greatly reduced, which seriously restricts the successful application of the reactive material in anti-light armor ammunition.’ to improve its comprehensibility. In lines 70-73.

  1. Comment::Line 123: replace “consists” with “consisting”.

Response: We replaced “consists” with “consisting”.

  1. Comment::Line 124: Check the sentence starting with "Put the solid ...". Probably replace "that placed" with "that is placed".

Response: We replaced "that placed" with "that is placed".

  1. Comment::Lines 207-209: the sentence starting with "Besides" is not clear. Please check and rephrase.

Response: We checked the sentence starting with "Besides" and rephrased for ‘The on-off signal is also the trigger signal for high-speed camera, overpressure transducer and pyrometer.’ in lines 211-212.

  1. Comment::Line 242: please replace the word "variate" with "variable".

Response: We replaced the word "variate" with "variable".

  1. Comment::Lines 303-304: The sentence starting with "Therefore" is incomplete. Please check and correct.

Response: The sentence starting with "Therefore" has been corrected for ‘Therefore, the PTFE based RMs are “frozen” in this case.’ in lines 318-319.

  1. Comment::Lines 315-318: the sentence starting with "The endothermic peak" is convoluted. Please rephrase and explain the contents better.

Response: The sentence starting with "The endothermic peak" has been corrected for “The endothermic peak B covers a temperature range from 530 ℃ to 601 ℃, at the same time, the mass of the samples decreases according to TG curve, indicating that small gas molecule is produced from the decomposition of PTFE matrix.” in lines 332-336.

  1. Comment::Lines 323-324: Authors write "leading to leading". Please revise to eliminate the almost consecutive repetition of the word "leading".

Response: “leading to” has been omitted.

  1. Comment::Lines 329-330: the sentence starting with "If the alumina layer ..." is incomplete. Please check and correct.

Response: The sentence starting with "If the alumina layer ..." has been corrected for “If the alumina layer is ruptured under the impact load, chemical reaction will bring forward to the decomposition temperature of PTFE.” in lines 347-349.

  1. Comment::Line 336: the adjective "exothermic" before the word "reaction" is superfluous. Please cancel it.

Response:  "exothermic" has been omitted.

  1. Comment::Line 361: The text reads "with that with". Please reprase to eliminate this repeat.

Response: “aluminum plate compared with that with” has been corrected for “aluminum plate as compared to that”.

  1. Comment::Line 398: correct "concluded" with "conclude".

Response: We corrected "concluded" with "conclude".

  1. Comment::Line 402: The sentence starting with "Integrate ..." is incomplete. It probably derives from a "cut and paste" operation. Please check and correct.

Response: The sentence starting with "Integrate ..." has been corrected for “Integrating the function of curves in Fig. 7, ” in lines 428-429.

  1. Comment::Line 22, line 406 and line 488: replace the word "introduce" with "introduction".

Response: We replaced the word "introduce" with "introduction".

  1. Comment::Lines 483-485: the sentence starting with "If the alumina ..." is incomplete. Please check and correct.

Response: The sentence starting with "If the alumina layer ..." has been corrected for “If the alumina layer is ruptured under the impact load, chemical reaction will bring forward to the decomposition temperature of PTFE.” in lines 510-512.

Reviewer 3 Report

The results of this manuscript are interesting. I would suggest a minor revision before publishing. Some suggestions and comments are listed below:

1- English of the manuscript needs polishing.

2- The main novelty of this work must be clearly mentioned as compared to other publications.

3- More physical interpretation about the experimental results can improve the quality of this work.

4- What are the limitations of your experiments and research data?

5- Some quantity results should be included in the abstract and conclusion. 

Author Response

Thanks very much for your attention on our manuscript (materials-1838343). We have made a careful revision on the revised manuscript according to the comments, and the revisions made to the manuscript have been marked up using the “Track Changes” function. The modification details are as follows:

  1. Comment::English of the manuscript needs polishing.

Response: We polished the English of the manuscript.

  1. Comment::The main novelty of this work must be clearly mentioned as compared to other publications.

Response: The main novelty of this work are as follows:

(1) The preparation and study of THV based reactive materials is an important feature of this paper.

(2) It is an innovation that the temperature and pressure of the product are taken into account when evaluating the energy release of the active material.

  1. Comment::More physical interpretation about the experimental results can improve the quality of this work.

Response: This paper focuses on the energy release efficiency of reactive materials with different formulations, and further analysis of the physical mechanism of the shock-induced energy release will be covered in the future work.

  1. Comment::What are the limitations of your experiments and research data?

Response: When comparing the energy release of different formulations of reactive materials, the speed should be the same or relatively close. Due to the limited test equipment and funds, the same impact speed is not obtained, which is one of the limitations of this paper.

  1. Comment::Some quantity results should be included in the abstract and conclusion. 

Response: We have added some quantity results in the abstract and conclusion.